# Dynamic expression of cathepsin L in the black soldier fly (*Hermetia illucens*) gut during *Escherichia coli* challenge

Yun-Ru Chiang[1,2], Han-Tso Lin[2], Chao-Wei Chang[2], Shih-Ming Lin[3], John Han-You Lin[1] *

**1** School of Veterinary Medicine, National Taiwan University, Taipei, Taiwan, **2** Department of Biotechnology, Ming Chuan University, Taoyuan, Taiwan, **3** Department of Biotechnology and Bioindustry Science, National Cheng Kung University, Tainan, Taiwan

* linhanyou@ntu.edu.tw

**Data Availability Statement:** All relevant data are within the paper.

**Funding:** The author(s) received no specific funding for this work.

## Abstract

The black soldier fly (BSF), *Hermetia illucens*, has the potential to serve as a valuable resource for waste bioconversion due to the ability of the larvae to thrive in a microbial-rich environment. Being an ecological decomposer, the survival of BSF larvae (BSFL) relies on developing an efficient defense system. Cathepsin L (CTSL) is a cysteine protease that plays roles in physiological and pathological processes. In this study, the full-length of *CTSL* was obtained from BSF. The 1,020-bp open reading frame encoded a preprotein of 339 amino acids with a predicted molecular weight of 32 kDa. The pro-domain contained the conserved ERFNIN, GNYD, and GCNGG motifs, which are all characteristic of CTSL. Homology revealed that the deduced amino acid sequence of BSF CTSL shared 74.22–72.99% identity with Diptera flies. Immunohistochemical (IHC) analysis showed the CTSL was predominantly localized in the gut, especially in the midgut. The mRNA expression of CTSL in different larval stages was analyzed by quantitative real-time PCR (RT-qPCR), which revealed that CTSL was expressed in the second to sixth instar, with the highest expression in the fifth instar. Following an immune challenge *in vivo* using *Escherichia coli* (*E. coli*), CTSL mRNA was significantly up-regulated at 6 h post-stimulation. The Z-Phe-Arg-AMC was gradually cleaved by the BSFL extract after 3 h post-stimulation. These results shed light on the potential role of CTSL in the defense mechanism that helps BSFL to survive against pathogens in a microbial-rich environment.

## Introduction

The black soldier fly (BSF), *Hermetia illucens* (Diptera: Stratiomyidae), populate tropical and temperate regions worldwide [1–4]. Its life cycle spans nearly forty days and comprises five distinct stages, including egg, larvae (from the first to the fifth instar), pre-pupa (the sixth instar), pupa, and adult [5–8]. It plays a crucial role by utilizing animal manure and organic waste as their primary food source in the larval stages. BSF larvae (BSFL) are now considered a valuable bio-converter with the ability to convert biomass into energy for their growth [9–16].

**Competing interests:** The authors have declared that no competing interests exist.

Cultivating BSFL offers several notable advantages. Firstly, they can transform various decaying plant and animal materials into valuable resources for animal feed and biofuel production [17–20]. Secondly, they represent an enticing opportunity as a potential substitute for conventional meal sources by virtue of their high protein and lipid content [21–23]. Thirdly, their ease of cultivation contributes significantly to environmental sustainability and the efficient recycling of resources [24]. These remarkable characteristics of BSFL align perfectly with sustainable development goals, particularly in promoting sustainable circular agriculture [25]. Owing to these BSFL advantages, they have become a commercially important species.

BSFL are utilized to convert low-grade food waste into valuable high-end proteins and fats within intensive production facilities; however, this could expose them to high concentrations of harmful microorganisms [26]. They are very plausible that this insect species has developed a heightened and more efficient immune response mechanism to cope with such environmental challenges [27]. In order to unravel their disease control mechanisms, develop effective disease control strategies, and optimize the yield of BSFL cultures, the complex immune mechanisms in these remarkable insects require in-depth study. Current knowledge regarding the immune response of BSFL indicates that a tough cuticular layer covers them to isolate themselves from their environment [28]. In their exposed digestive tract, salivary glands and digestive organs such as the midgut secrete digestive fluids to break down microorganisms from their food. Upon the invasion of microorganisms into the body, blood cells or fat body cells secrete soluble protein molecules, including lectins [29], immune peptides, and antimicrobial peptides [30–34]. These bioactive molecules play a crucial role in combating the invading pathogens [35–37]. This process leads to nodule formation and encapsulation responses to inhibit the spread of infections or direct phagocytosis of pathogens by blood cells [38–40]. Despite appearing to adapt to environments rich in microorganisms, some reports indicate that BSFL can also be susceptible to various entomopathogens that cause damage, including fungi, viruses, protozoa, and bacteria [26, 41]. Therefore, a thorough understanding of the immune system of BSFL will better equip us to utilize them in large-scale applications for environmental sustainability and provide insights into disease control strategies.

Cathepsin L (CTSL) is a crucial cysteine protease widely present across various organisms, including insects [42–46]. It plays a pivotal role in numerous facets of insect biology, thus drawing significant attention from entomologists. This enzyme exhibits a remarkable involvement in a variety of life processes, including molting [43], the catabolism of fat bodies [44], programmed cell death [44], wing disc differentiation [47], eclosion [48], metamorphosis [49, 50], and immune responses [51]. In the context of insects, CTSL is predominantly localized within the digestive tract, specifically the midgut [51]. It is worth noting that CTSL is also present in other organs, such as in the testis, head, midgut, and hemocytes [52, 53]. Notably, its involvement in the degradation of foreign pathogens is crucial within the scope of the insect immune response, where the primary function of CTSL involves food digestion and the prevention of pathogenic invasion of the digestive tract [51]. CTSL also demonstrates antibacterial activity [35], particularly in cases of digestive tract invasion, potentially influencing various facets of innate immune responses [54, 55]. Nevertheless, the specific expression patterns of CTSL in BSFL and its precise role in microbial infection remain intriguing areas of inquiry that await further investigation from the perspective of entomological research.

In this study, our aim was to elucidate the molecular characteristics of CTSL in BSFL. We took a multifaceted approach and integrated various approaches. First, we obtained the CTSL gene in BSFL using rapid amplification of cDNA ends (RACE) by designing degenerate primers derived from different Diptera strains. These primers helped to obtain partial CTSL sequences. Then, we performed RACE to gain the complete open reading frame of the CTSL sequence in BSFL. Second, hematoxylin and eosin (H&E) staining was used to gain a

comprehensive understanding of the overall tissue structure and morphology of BSFL, and in parallel, we generated anti-CTSL serum to precisely visualize CTSL localization within BSFL tissue by immunohistochemistry (IHC) analysis. Third, we performed CTSL mRNA expression profiling to thoroughly examine CTSL mRNA expression levels in various larval stages of BSFL. This process provided essential insights into the dynamic regulatory patterns of CTSL at different developmental stages. At last, a microbial challenge experiment was conducted in which pathogen invasion *in vivo* was simulated. We challenged BSFL with *Escherichia coli* (*E. coli*) by physically inducing damage and then explored the changes in CTSL mRNA expression levels and proteolytic activity. The mRNA expression levels were assessed at multiple time points using quantitative real-time PCR (RT-qPCR), and proteolytic activity was quantified using the Z-Phe-Arg-AMC.

Our comprehensive investigation greatly enhanced the understanding of CTSL in BSFL. Furthermore, we revealed the dynamic changes in CTSL expression during microbial infection, revealing the critical role of CTSL in the BSFL immune response. Given the great significance of BSFL in the context of the circular economy, an in-depth exploration of its immunology is crucial. This research has provided further insight into our knowledge of entomology. It has the potential to drive advancements in industry and practices rooted in BSFL by which to achieve sustainability and ecological efficiency.

## Materials and methods

### Insect rearing

BSFL were supplied by Prof. Han-Tso Lin's lab at Ming Chuan University, Taiwan. BSFL were reared in a 5-liter plastic bucket with 500 g of wheat bran on the bottom as bedding material and feed. The environment was maintained at 27±1°C in a 16 h photoperiod at 60–70% relative humidity [56]. BSFL were divided into six instar stages based on their characteristics and lengths, according to the report by Barros [8]. The second to sixth instar were selected in each group for the following experiments, which were performed in triplicate.

### RNA isolation and cDNA synthesis from BSFL

Total RNA extraction was performed for cDNA sequence and mRNA expression analysis. Fresh BSFL (5–10 mg body weight) were snap-frozen in liquid nitrogen and then homogenized and frozen by mortar and pestle. Total RNA was extracted using TRIzol reagent (Invitrogen, CA, USA) and GENEzol TriRNA Pure Kit (Geneaid Biotech, Taiwan) with an additional chloroform extraction step. RNA was eluted in 50 µl of RNase-free double distilled water; the quality, concentration, and purity (the absorbance ratios in 260/280 nm and 260/230 nm) were analyzed by measuring the absorbance using a spectrophotometer (Synergy HT, BioTek Instruments, VT, USA). Total RNA (200 ng) was used directly as a template for cDNA synthesis, in which the RNA samples were reverse transcribed using M-MLV reverse transcriptase (M-MLV RT) (Promega Corp, Madison, WI, USA) and oligo(dT)-primers (Promega Corp, Madison, WI, USA). The reaction was incubated at 42°C for 60 min and then at 95°C for 15 min. The final synthesized cDNA was stored at -80°C for subsequent use.

### Obtaining the total open reading frame of the CTSL cDNA sequence from BSFL

Degenerated primers (Table 1) were utilized to amplify a partial *cathepsin L* gene cDNA. Two pairs of degenerated primers were designed on the conserved cDNA sequences of CTSL from six species of Diptera flies, including *Drosophila bipectinata* (XM_017251151.1),

**Table 1. Primers for RACE and quantitative real-time PCR used in this study.**

| Primer name | Primer sequence (5'-3') |
| --- | --- |
| Degenerate primers | |
| CTSL-1F | AGAACAAGCACAARATYGCCAAG |
| CTSL-1R | YTANACYARBGGRTARCTGG |
| CTSL-2F | ABGAYYWVGKDYMYTGCKGC |
| CTSL-2R | CCANGAGTTVBYMAYYARCC |
| RT-PCR primers | |
| qCTSL-F | GGGTGCTGTGACCCCAATAA |
| qCTSL-R | CCATAAGACCGCCATTGCAT |
| Actin-F | ACGTTGCAATCCAGGCTGTT |
| Actin-R | CACGAACGATTTCCCTTTCG |

*Drosophila eugracilis* (XM_017213434.1), *Drosophila serrata* (XM_020945285.1), *Anopheles gambiae* (XM_001689230.2), *Aedes aegypti* (XM_001655949.2), and *Culex quinquefasciatus* (XM_001848292.1). These CTSL cDNA sequences were aligned by Clustal Omega (https://www.ebi.ac.uk/Tools/msa/clustalo). The conserved sequences were selected to represent the wide spectrum of phylogenetic diversity in CTSL. PCR was performed in a 30 μl volume with 2 mM of 2X Perfect Read PCR Master Mix (Ten Giga Bio, Taiwan), cDNA (200 ng), and degenerate primers (1 mM). PCR amplification conditions for the partial sequence were as follows. The cycling parameters included a 95˚C hot start for 5 s, followed by 35 cycles of a three-step PCR: 95˚C for 30 s, an annealing temperature of 53˚C for 30 s, and a 72˚C extension step for 45 s, with a final extension at 72˚C for 10 s. The PCR products were cloned into pGEM®-T Easy Vector Systems (Promega Corp, Madison, WI, USA). Positive clones with the expected inserts were sequenced using T7 and SP6 primers. After receiving the partial *cathepsin L* sequence, the full-length cDNA sequence of the open reading frame in *cathepsin L* was obtained through the RACE technique (GeneRacer™ Kit, Invitrogen). Primers were designed based on the obtained partial sequence, and the process was performed following the manufacturer's instructions. After RACE, the full-length open reading frame of the BSFL *cathepsin L* gene was obtained, and the sequence was analyzed by Mission Biotech (Taipei, Taiwan).

## Phylogenic analysis of CTSL amino acid sequence

The obtained BSFL *cathepsin L* sequence was translated via EXPASY (https://web.expasy.org/translate) to deduce its amino acid sequence. Signal peptides were predicted by SignalP-6.0 (https://services.healthtech.dtu.dk/service.php?SignalP). The deduced amino acid sequence was aligned via Clustal Omega (https://www.ebi.ac.uk/Tools/msa/clustalo). Percent identity and a phylogenetic tree were calculated using MEGA11 software (https://www.megasoftware.net) adopting the neighbor-joining algorithm with available insect, arthropod, fish, and mammal CTSL amino acid sequences obtained from GenBank with the following accession numbers: D*rosophila bipectinata* (XP_017106640.1), *Drosophila eugracilis* (XP_017068923.1), *Drosophila serrata* (XP_020800944.1), *Anopheles gambiae* (XP_001689282.1), *Aedes aegypti* (XP_001655999.2), *Culex quinquefasciatus* (XP_001848344.1), *Penaeus monodon* (AME17651.1), *Macrobrachium rosenbergii* (AJP62583.2), *Danio rerio* (CAA69623.1), and *Homo sapiens* (NP_001244901.1). Phylogenetic trees were produced by bootstrapping this matrix with 1,000 replicates.

## Morphological organization of BSFL and anti-CTSL serum preparation for IHC analysis

For H&E and IHC staining, fresh tissues from the fifth instar were fixed in Davidson's alcohol formalin acetic acid fixative and stored at room temperature for 24 h. Fixed tissues were dehydrated using increasing concentrations of ethanol (70%, 80%, 90%, 95%, and 100%), cleared twice in xylene, and then embedded in melted paraffin. Sections (4 μm) cut using a microtome (Accu-Cut SRM 200 Rotary Microtome, Sakura Finetek, CA, USA) were mounted on glass slides, deparaffinized in xylene, rehydrated in decreasing concentrations of ethanol (100%, 95%, 90%, 80%, and 70%), and washed in ddH$_2$O. H&E staining was performed by immersing the slides in Mayer's hematoxylin (Sigma-Aldrich) with agitation for 30 s, followed by a 2 min rinse under running tap water. The slides were then stained with a 1% eosin solution for 30 s. Subsequently, the sections underwent two rounds of dehydration in 95% alcohol and two rounds in 100% alcohol (each for 30 s) before being cleared in xylene. Finally, the slides were treated with mounting medium and covered with coverslips. Recombinant CTSL protein was produced and purified from *E. coli* BL21 (DE3) for IHC staining following the methods outlined in our previous study [57]. SDS-PAGE was used to identify recombinant CTSL, presumed at a molecular weight of 32 kDa. Purified recombinant CTSL protein was used as an antigen for CTSL antiserum production in BALB/c mice (S1 Fig). Equal volumes of an antigen solution (CTSL 0.5 mg/ml in phosphate-buffered saline, PBS) were mixed with Freund's complete adjuvant (1:1, v/v) for the primary injection. The mice were subcutaneously injected with 0.5 ml of the antigen mixture, followed by a booster injection with the same antigen mixture in the fourth week. Serum was collected from the hearts of the experimental mice, clotting at 25°C for 2 h, and centrifuged at 3,000 × g at 4°C for 10 min. Purified mice anti-CTSL polyclonal antibodies were then isolated. An enzyme-linked immunosorbent assay was used to determine antibody titers for the anti-CTSL antibodies (S2 Fig). For IHC analysis, BSFL tissue sections were incubated with anti-CTSL primary antibodies (1:500, v/v) in PBS at room temperature for 30 min. The sections were washed twice using PBS and incubated with One-Step enzymes horseradish peroxidase polymer for 20 min, then stained in a chromogen substrate diaminobenzidine for 10 min. Sections were subsequently counterstained in Mayer's hematoxylin (Sigma-Aldrich) for 2 min, dehydrated using ethanol, cleared in xylene, and mounted.

## Analysis of BSFL CTSL mRNA expression in different instars

BSFL were freshly collected in triplicate at different instar phases, specifically the second to sixth instars, and homogenized for RNA extraction. Gene expression was analyzed via RT-qPCR using SYBR Green qPCR MasterMix (Promega Corp, Madison, WI, USA). Primer sequences are listed in Table 1. The Mx3005P qPCR system (Agilent Technologies, CA, USA) performed amplification, detection, and analysis. The amplification program consisted of preincubation at 95°C for 3 min, followed by 40 cycles of denaturation at 95°C for 10 s and annealing at 60°C for 60 s. Expression levels of β-actin (β-actin-F/R) mRNA served as an internal control. The relative mRNA levels are presented as $2^{-\Delta\Delta ct}$ [58].

## Dynamic expression of CTSL mRNA in response to *E. coli* challenge

To assess dynamic CTSL expression under challenge, samples ranging from the second to sixth instar were evaluated. The CTSL profile under *E. coli*-stimulated conditions was determined in triplicate, and the *E. coli* bacterial strain K12 (Sigma-Aldrich, USA) was used for all experiments. Each BSFL was pricked with a fine needle, and then dipped into a suspension of *E. coli* (OD$_{600\ nm}$ = 1.6, approximately 2 X 10$^9$ CFU/ ml) for 15 min. After *E. coli* exposure, the

larvae were transferred into Petri dishes and incubated for 1.5, 3, 6, 9, and 12 h before being collected, homogenized, and frozen by mortar and pestle. The resulting homogenates were used for RNA extraction, cDNA synthesis, and RT-qPCR detection. The control groups were not subjected to any form of treatment.

## Measurement of proteolytic activity from the BSFL extract after *E. coli* challenge

Needles wounded the fifth instars and then immersed in a suspension of *E. coli* ($OD_{600 \, nm}$ = 1.6, approximately 2 X $10^9$ CFU/ ml) for 15 min. As previously described, samples were collected after 1.5, 3, 6, 9, and 12 h. Each time course was repeated five times. Samples were homogenized in a PBS solution (0.01 M, 1 ml) in a grinder on ice. CTSL enzyme activity was quantified according to Werle's report [59] with slight modifications outlined in our previous report [57]. BSFL were added to PBS, homogenized, and centrifuged at 4°C at 14,000 g for 30 min. A 50 μl aliquot of the supernatant was mixed with 100 μl of 0.1 M sodium acetate buffer (NaOAc/acetic acid buffer, pH 5.0) and 50 μl of 10 μM *Z*-Phe-Arg-amido-4-methyl coumarin hydrochloride (*Z*-Phe-Arg-AMC; Sigma-Aldrich, Saint Louis, USA), which is a cysteine protease-specific synthetic fluorometric substrate [60, 61]. Cysteine protease can cleave the substrate by attacking the peptide bond between arginine and AMC. After the release of AMC, we detected its fluorescence using a microplate fluorometer at an excitation wavelength of 380 nm and an emission wavelength of 450 nm (Synergy HT, BioTek Instruments, Winooski, VT). L-trans-Epoxysuccinyl-leucylamido (4-guanidino) butane (E-64, 10 μM), an inhibitor of cysteine proteinases, was a negative control. The relative activity was calculated using the following formula: Relative activity = fluorescence in the treatment group / the highest fluorescence in the trial.

## Statistical analysis

Statistical analysis in this study was conducted by one-way analysis of variance (ANOVA) and Dunnett's test using Graph-Pad Prism 8.4 for Mac (GraphPad Software, CA, USA), where $p < 0.05$ was considered statistically significant between treatment and control groups.

## Results

### Nucleotide and deduced amino acid sequence of BSFL CTSL

The full-length of the *CTSL* open reading frame was 1,020 bp and encoded a polypeptide of 339 amino acids (a.a.) with a predicted molecular weight of 32 kDa (Fig 1). The deduced amino acids included a pre-domain of CTSL from 1–16 a.a., a pro-domain from 17–121 a.a., and a catalytic domain from 122–337 a.a. The putative signal peptide was 16 a.a. from Met[1] to Ala[16]. The putative mature domain was from Pro[122] to Pro[337]. The catalytic triad was formed from Cys[146], His[285], and Asn[306]. Five S2 subsites were defined by the conserved residues Leu[190], Met[191], Ala[257], Leu[283], and Gly[286], and three conserved disulfide bridges were formed between Cys[143] to Cys[186], Cys[177] to Cys[219], and Cys[278] to Cys[328]. The pro-domain contained three CTSL-specific motifs, including an ERFNIN motif from 44–62 a.a., a GNYD motif from 74–81 a.a., and a GCNGG motif from 67–80 a.a. (Fig 2). Based on the above, it was inferred that the sequence obtained in this study was indeed encoded for CTSL.

### Phylogenetic analysis of CTSL

A multiple sequence alignment showed that the deduced amino acid sequence of CTSL from BSFL was highly homologous to its counterpart Diptera species (Fig 3). Phylogenetic analysis

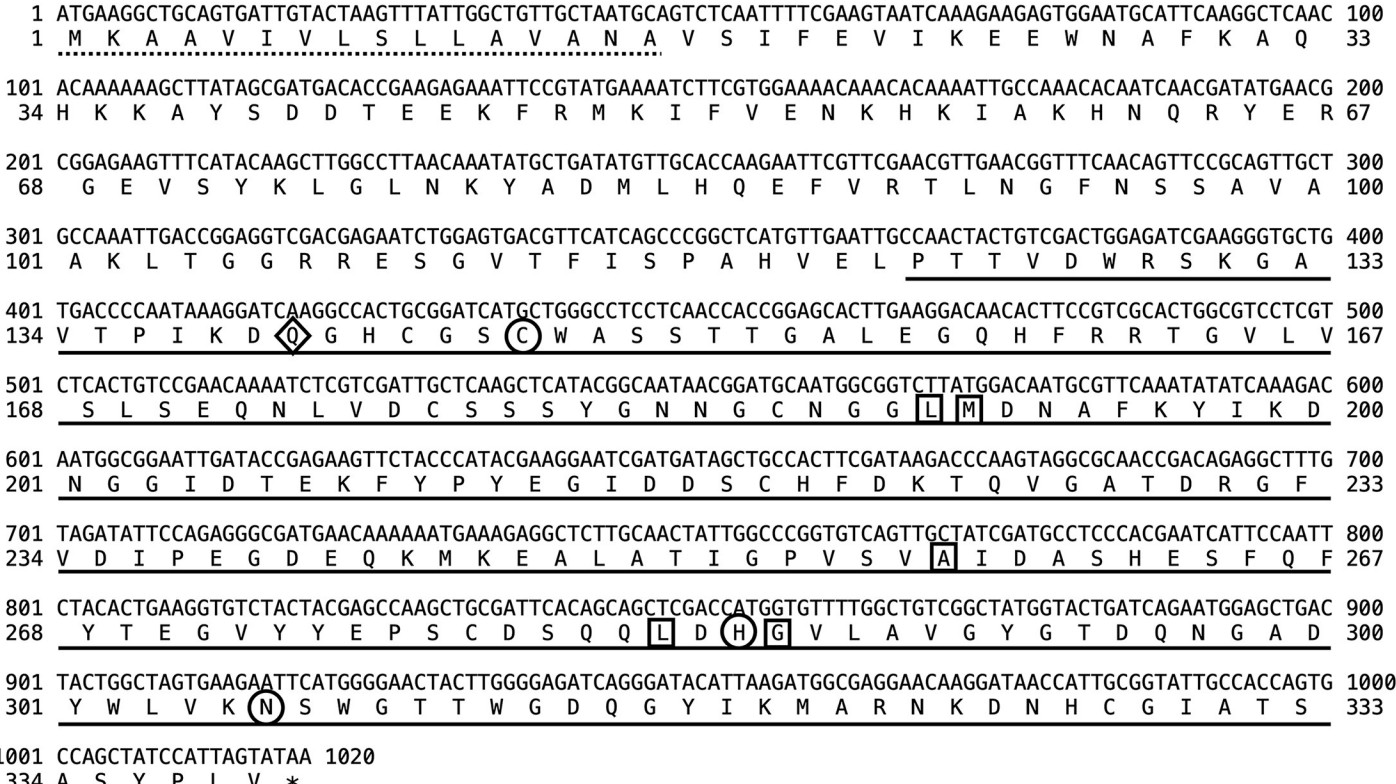

**Fig 1. Nucleotide and deduced amino acid sequence of BSFL CTSL.** The dotted line indicates the putative signal peptide sequence (Met[1]-Ala[16]), and the putative mature domain is underlined (Pro[175]-Pro[337]). The triad of conserved catalytic active sites (Cys[146], His[285], and Asn[306]) is in circled frames, and Gln[140] of the oxyanion hole is in the diamond frame. Five S2 subsites are defined by residues Leu[190], Met[191], Ala[257], Leu[283], and Gly[286] and indicated by the square frames. The asterisk marks the stop codon at the end of the open reading frame.

indicated that the deduced amino acid sequence of BSFL CTSL shared 74.22–72.99% homology with Diptera flies. Overall homologies with other Diptera flies were as follows: the highest identity of 74.22% with *D. serrata*, 72.99% with *D. bipectinate*, 73.15% with *D. eugracilis*, 72.57% with *A. aegypti*, 71.14% with *A. gambiae*, 71.76% with *C. quinquefasciatus*, 63.80% with *M. rosenbergii*, 66.77% with *P. monodon*, 51.02% with zebrafish *D. rerio*, and 50.29% with human *H. sapiens*. It was also found that BSFL CTSL was grouped with the CTSL from arthropods and was closer to the family Drosophilidae than to Culicidae.

## Morphological organization of BSFL and localization of CTSL in BSFL

In the microstructure observation of BSFL, H&E staining revealed the main structure of the fifth instar in the longitudinal and cross sections. Our results showed that the gut was the major structure in the body cavity, surrounded by a muscular layer and covered by a chitin outer skin. The major section is the midgut (Fig 4A and 4D). To obtain a more precise profile of CTSL localization in BSFL, IHC was used to analyze the distribution of CTSL proteins in BSFL organs. Positive signals were detected as a brownish color following the addition of a chromogenic agent. This agent reacted with the enzyme carried by antiserum against BSFL CTSL in the antiserum group (Fig 4C and 4F). The entire midgut was positive for the BSFL CTSL antiserum, indicating the presence of CTSL in the gut of BSFL. Contrastingly, there was

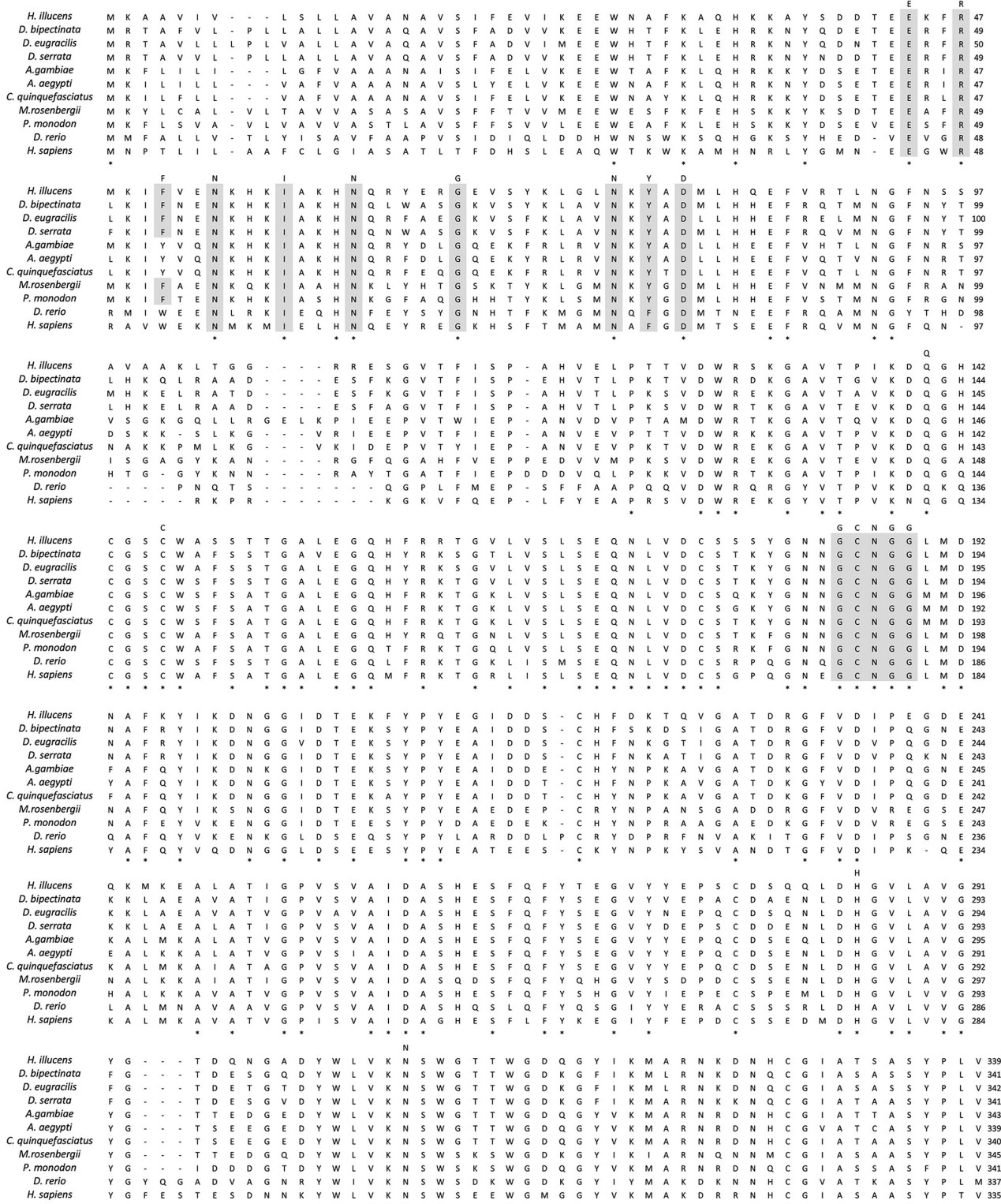

**Fig 2. Alignment of amino acid sequences of CTSL from representative arthropods and vertebrates.** Asterisks (*) indicate amino acids identical in all species. Dashes (–) denote gaps. The CTSL family signatures, including ERFNIN, GNYD, and GCNGG are shaded.

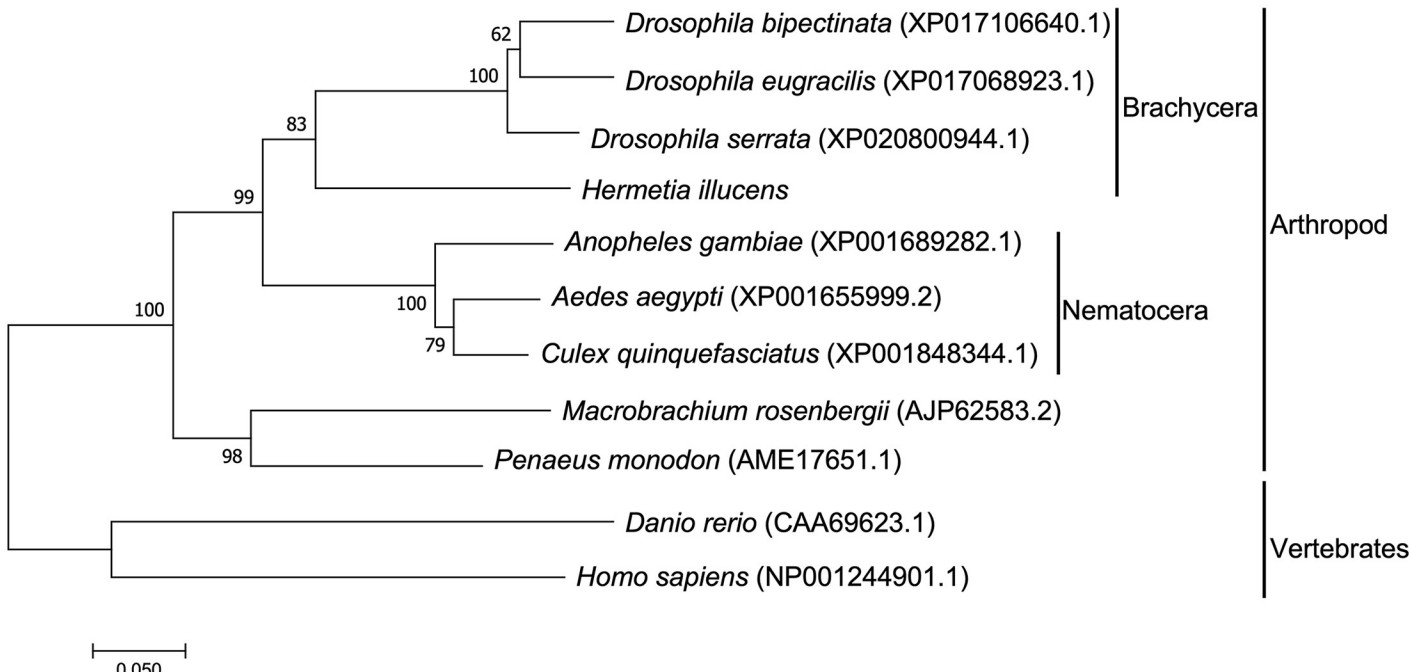

**Fig 3. Unrooted phylogenetic tree of CTSL and other homologs.** The neighbor-joining algorithm constructed the tree using the Mega11 program based on the multiple sequence alignments by Clustal Omega. The branches indicate bootstrap values of 1,000 replicates (%). The scale bar corresponds to 0.05 estimated amino acid substitutions per site. The amino acid sequences of all species were obtained from the NCBI database. GenBank accession numbers are as follows: *Drosophila bipectinata* (XP017106640.1), *Drosophila eugracilis* (XP017068923.1), *Drosophila serrata* (XP020800944.1), *Anopheles gambiae* (XP001689282.1), *Aedes aegypti* (XP001655999.2), *Culex quinquefasciatus* (XP001848344.1), *Macrobrachium rosenbergii* (AJP62583.2), *Penaeus monodon* (AME17651.1), *Danio rerio* (CAA69623.1), and *Homo sapiens* (NP001244901.1).

no signal detected in the body cavity, muscle, or outer skin positions. Likewise, no signal was observed for the naïve serum employed as a negative control (Fig 4B and 4E).

## Profile of CTSL mRNA expression in different instars of BSFL

To quantify the expression of CTSL in various BSFL instars, we determined the expression profiles of the CTSL gene. The mRNA expression of CTSL in the second instar served as a relative control of 1.00. CTSL mRNA expression was as follows: the third instar was 0.95, the fourth instar was 1.30, the fifth instar was 6.86, and the sixth instar was 1.64. Thus, the highest level of CTSL was expressed in the fifth instar (Fig 5).

## Dynamic expression of CTSL mRNA in response to *E. coli* challenge

A time course of temporal CTSL mRNA expression was performed following *E. coli* challenge. The results showed that CTSL in all instars gradually increased after 3 h post-stimulation and dramatically increased at 6 h post-stimulation. The expression of CTSL in the second, third, fourth, and sixth instars was 5 times higher than that of the control, whereas the expression in the fifth instar was 13 times higher than that of the control (Fig 6). After 6 h post-stimulation, the expression level of CTSL gradually decreased and returned to control levels.

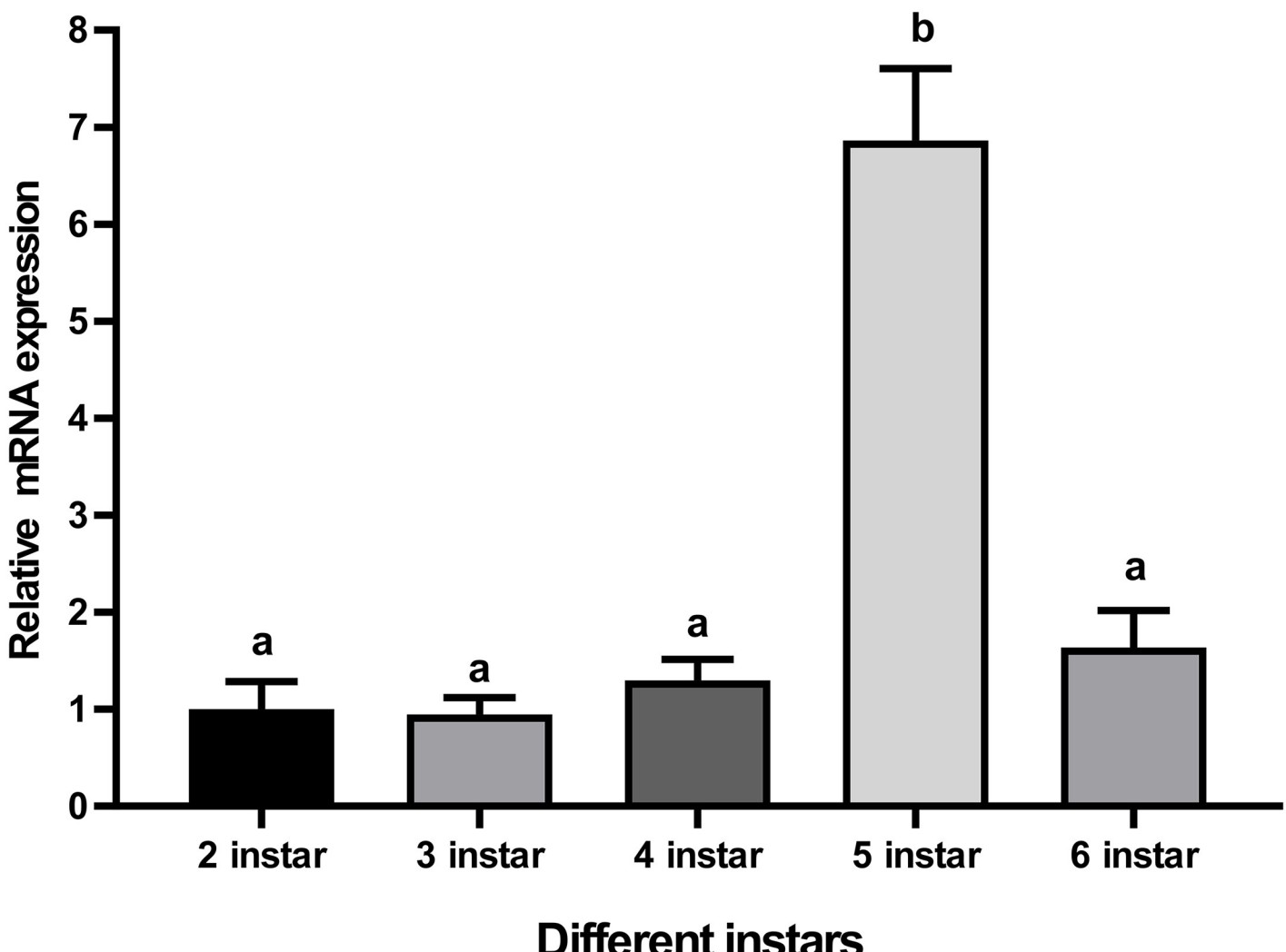

**Fig 4. Morphological organization and immunohistochemical localization of CTSL in BSFL.** The histology of BSFL is presented in longitudinal sections (A, B, C) and cross sections (D, E, F). H&E staining revealed the main structure of the fifth instar of BSFL(A, D). BSFL were stained with naïve mouse serum as a negative control (B, E). BSFL were stained by chromogenic IHC methods following injection of mice anti-BSFL CTSL serum (C, F). Positive signals were revealed in brown, while hematoxylin was used as a counterstain. H, head. (Bar = 500 μm).

## Dynamic proteolytic activity of cysteine cathepsins in response to *E. coli* challenge

To measure the effect of infection on the activity of cysteine cathepsins from BSFL, a proteolytic assay was conducted on the fifth instar of BSFL at 0, 1.5, 3, 6, 9, and 12 h post-infection (Fig 7). The proteolytic activity of cysteine cathepsins exhibited a gradual increase from 3 to 12 h after the *E. coli* challenge, reaching its peak at 12 h post-stimulation. In the negative control, the relative activity of cysteine cathepsins decreased by only 20% in 12 h in samples incubated with the cysteine proteinase inhibitor E-64.

## Discussion

This study marks pioneering work in identifying, sequencing, and characterizing the BSFL *CTSL*. The length of *CTSL* in organisms from vertebrates to arthropods typically ranges from

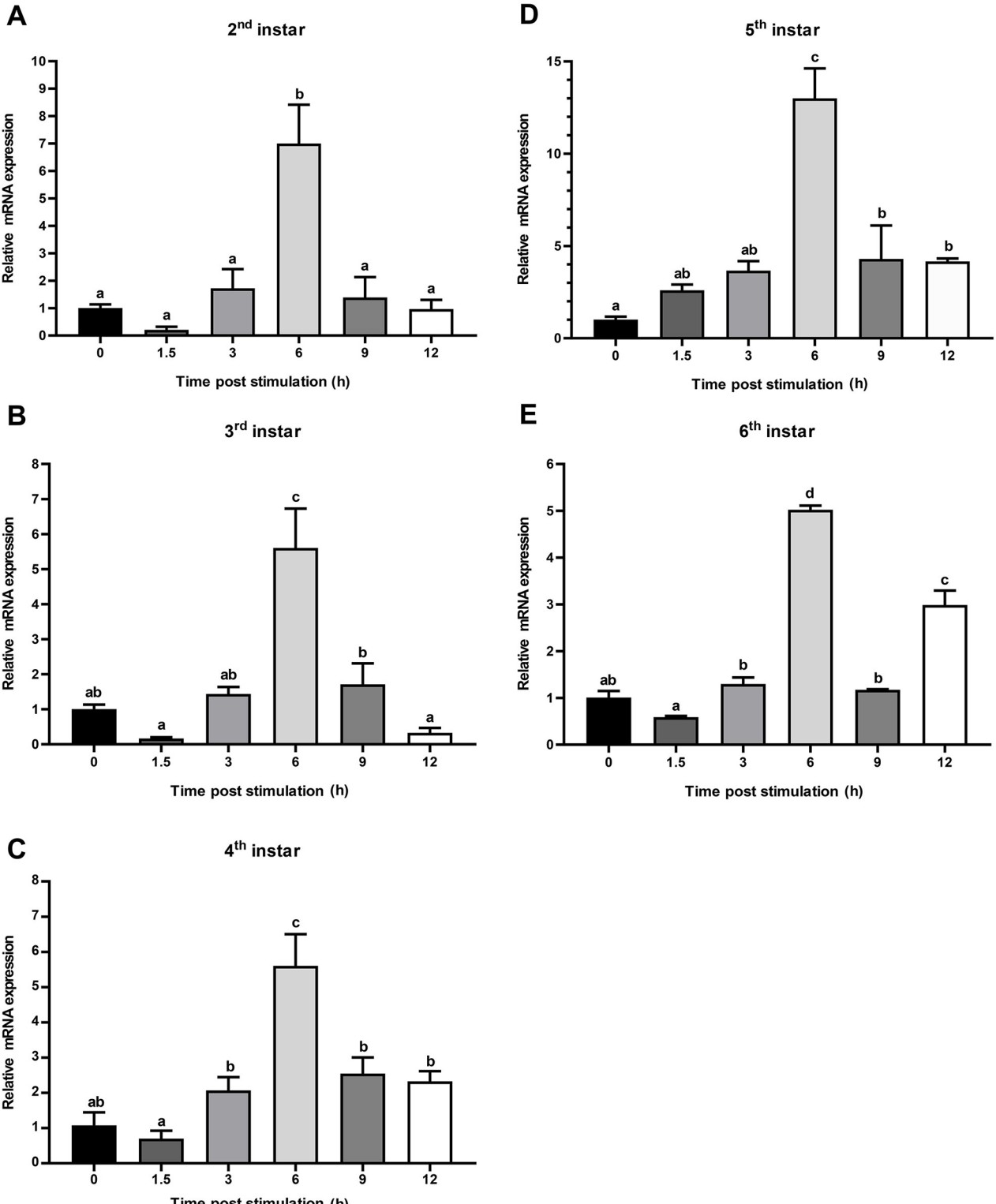

**Fig 5. Relative expression levels of CTSL mRNA in the different larval stages.** Relative mRNA expression ($2^{-\Delta\Delta ct}$) levels of CTSL were detected using SRBR green assay and normalized to β-actin. The fold-change of CTSL expression was calculated for the 3rd, 4th, 5th, and 6th instars and compared with the 2nd instar. Bars represent mean ± S.E. (n = 3). Letters indicate significant differences ($p < 0.05$).

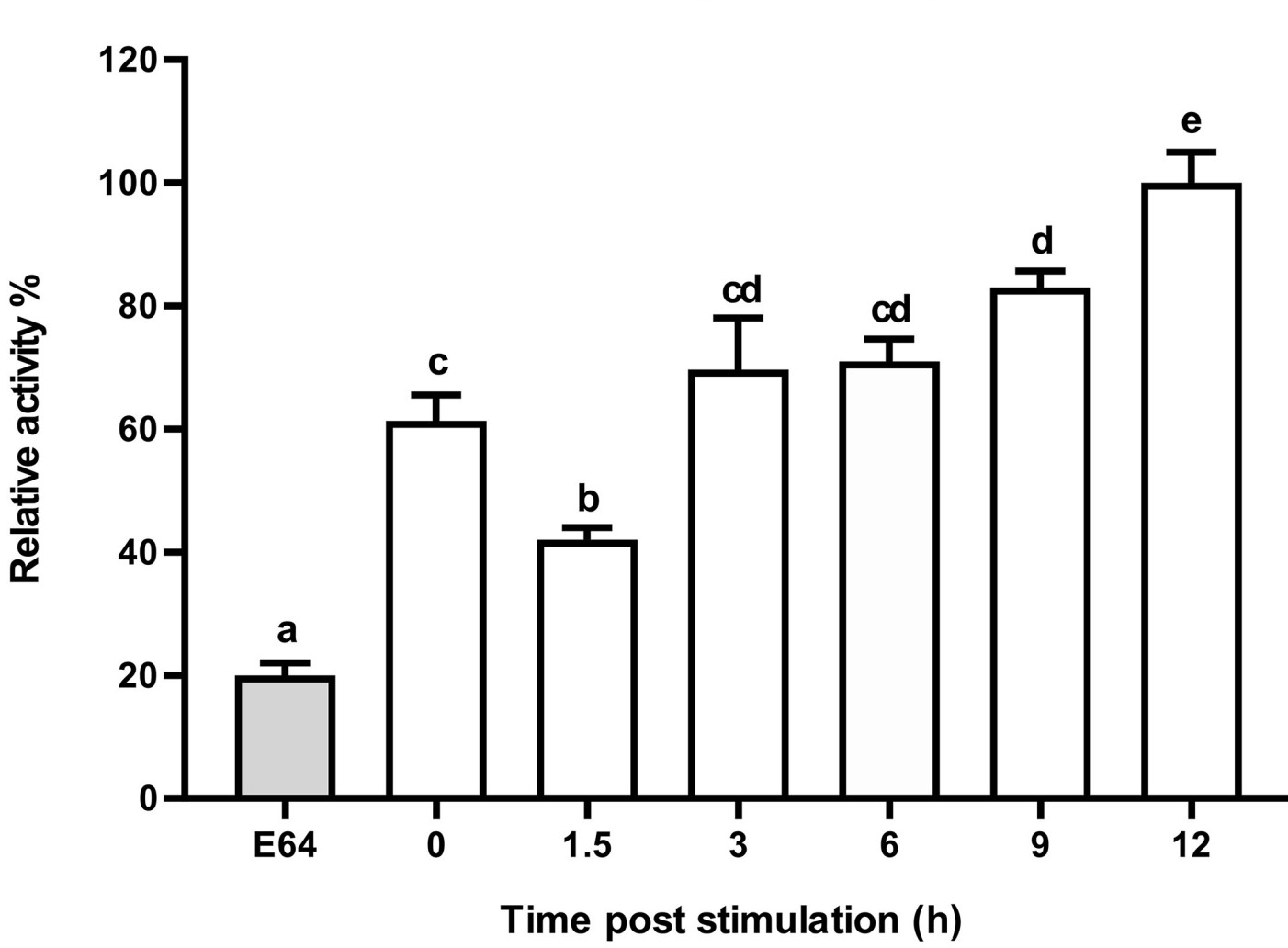

**Fig 6. Dynamic expression of CTSL mRNA in response to *E. coli* challenge.** Relative mRNA expression ($2^{-\Delta\Delta ct}$) levels of CTSL were detected using SRBR green assay and normalized to β-actin in the 2nd (A), 3rd (B), 4th (C), 5th (D), and 6th (E) instars. The fold-change of CTSL expression was calculated at 1.5, 3, 6, 9, and 12 h post-stimulation with *E. coli* and compared with 0 h (negative control was non-treated BSFL). Bars represent mean ± S.E. (n = 3). Statistical significance is indicated with lowercase letters ($p < 0.05$).

335 to 345 a.a. The BSFL CTSL generated in our study at 339 a.a. was consistent with that of other organisms. These similarities in sequence length emphasize the conservation of CTSL among different species. Note that variations in CTSL sequences exist between species and that CTSL recognition relies on unique protein motifs. The characteristic motifs ERFNIN, GNYD, and GCNGG are conserved features of CTSL in these species, representing the catalytic domain of cysteine proteases within the peptide C1A domain, which are also known as papain-like enzymes [46].

Characterized by the "R" residue, the uniqueness of the ERFNIN motif is a distinct feature found in Brachycera BSFL. In contrast, the EYFNIN and EWFNIN motifs are commonly found in nematodes and vertebrates, respectively. These changes in characteristic patterns may be attributed to evolutionary differences between species. Phylogenetic analysis further

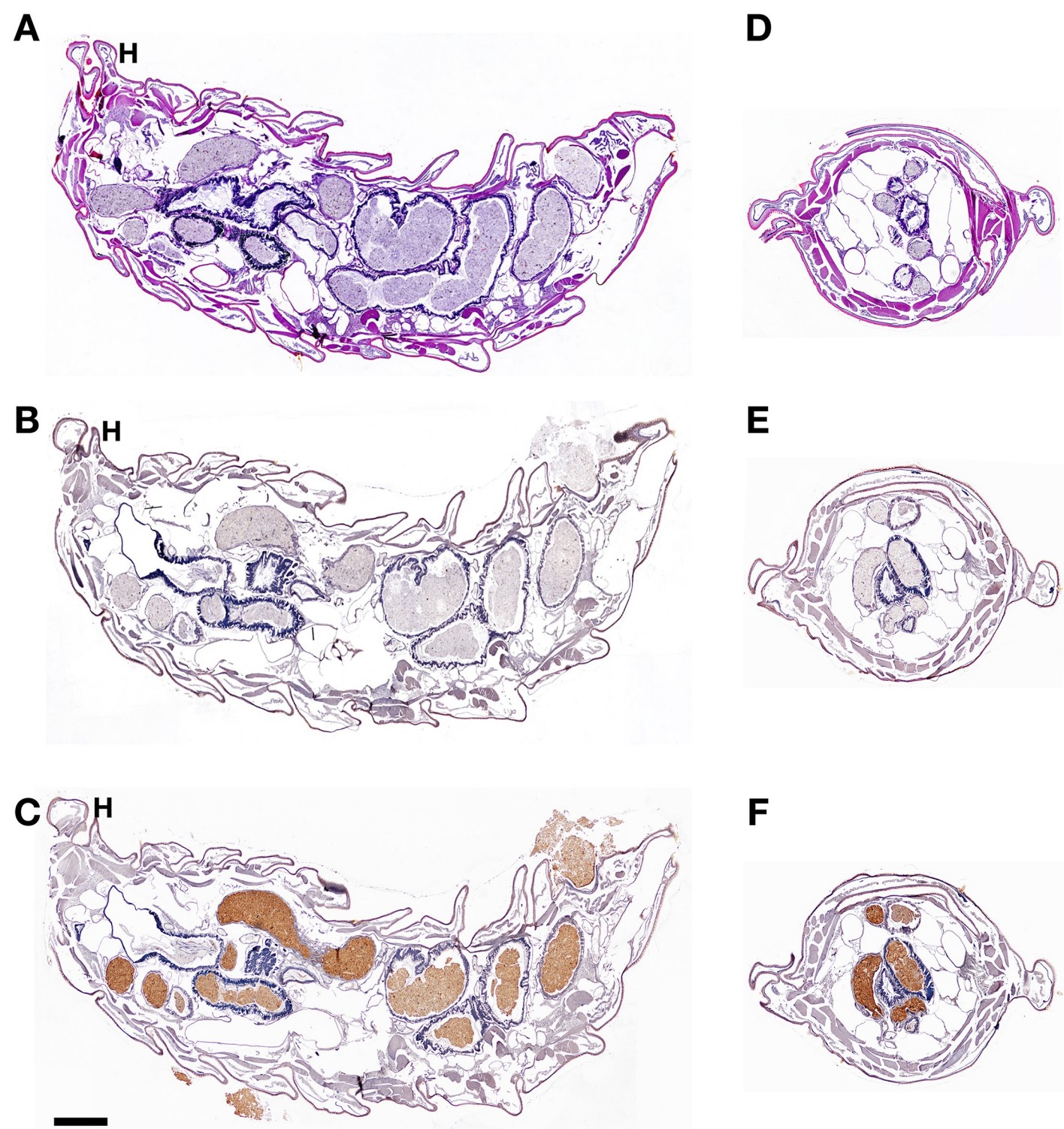

**Fig 7. Dynamic proteolytic activity of CTSL in response to *E. coli* challenge.** Percentage of CTSL activity in the fifth instar under *E. coli* challenge compared to negative control (cysteine proteinase inhibitor E-64). Error bars represent mean ± S.E. (n = 5). Statistical significance is indicated with different lowercase letters ($p < 0.05$).

showed that CTSL in BSFL is closely related to Brachycera in Diptera, suggesting a potential functional homology with Brachycera.

The tissue distribution of CTSL in organisms often corresponds to their specific functions. In our study, we have provided insights into the localization of CTSL in BSFL. Our results revealed that CTSL was predominantly expressed in the intestine, with a strong IHC positive signal observed in the midgut. As the main site for nutrient digestion and absorption, the midgut in BSFL has unique luminal pH changes and specific morpho-functional characteristics [62, 63]. It also possesses a strongly acidic pH region with high enzymatic activity, suggesting a crucial role in defense by effectively eliminating pathogens ingested with the feeding substrate [51, 62, 64–66]. This is consistent with the presence of CTSL in the midgut, as observed in some other insects, such as *Acyrthosiphon pisum*, where CTSL plays a crucial role in food digestion and regulating immune responses during infection [67]. Moreover, CTSL purified from the midgut of *Riptortus pedestris* has potent antibacterial activity against *Burkholderia* [54].

Our study also revealed dynamic changes in CTSL expression at different stages of BSFL, with the highest expression levels observed in the fifth instar. This pattern is consistent with that in *Bombyx mori*, where CTSL exhibits different expression levels across larval instars, peaking at the fourth instar [68]. In *Helicoverpa armigera*, CTSL proteolytic activity is significantly higher during the fifth to sixth instar molt [43], and injections of CTSL inhibitors resulted in delayed molting from the fifth instar to the sixth instar. Interestingly, the fifth instar BSFL possesses an efficient defense system against infection [28]. This may explain the increased expression of CTSL in the fifth instar, suggesting a crucial role in defense mechanisms and molting at this developmental stage [28, 43]. The exact function of CTSL in BSFL requires further investigation.

In this study, we simulated pathogen invasion and examined the expression of CTSL under various conditions. Our observations indicate the rapid appearance of melanized spots on the cuticle after challenge with *E. coli*, coupled with a significant increase in CTSL mRNA expression levels after stimulation in all larval stages. This profile parallels the observed proteolytic activity assay. The Z-Phe-Arg-AMC was gradually cleaved by the BSFL extract after 3 h of *E. coli* stimulation. Although it was challenging to prove that Z-Phe-Arg-AMC is exclusively cleaved by CTSL, our observations suggest that CTSL abundant expression in the body. Therefore, CTSL might play a crucial role in the trial involving the cleavage of Z-Phe-Arg-AMC. However, further investigation is required to substantiate this perspective. This dynamic response suggests that CTSL levels fluctuate in response to environmental stimuli and help BSFL to better adapt to microbial-rich environments. CTSL not only passively degrades microorganisms in food but may also contribute to the regulation of the innate immune response. In addition to antibacterial activity [54, 57], CTSL also can modulate various immune pathways in insects. In *citrus psyllids*, knockdown experiments targeting CTSL led to substantial modifications within innate immune pathways, encompassing Toll, MyD88, IMD, Relish, Dorsal, Cactus, and FADD [55]. CTSL has been shown to play a crucial role in regulating innate immune responses through pathways involving Toll, MyD88, Tube, and Pelle.

## Conclusion

We investigated the characteristics of CTSL in BSFL by cloning the CTSL gene using degenerate primers. CTSL emerged as a predominant molecule within the gut, primarily expressed in the midgut. Its presence was noticeable throughout different instars, with the highest mRNA expression observed during the fifth instar. Following *E. coli* stimulation, CTSL expression could be induced at the mRNA level, and the Z-Phe-Arg-AMC was cleaved by the BSFL

extract. These results suggest that treating BSFL with CTSL may effectively manage diseases, boost BSFL immunity, and contribute to sustainability in circular agriculture.

## Supporting information

**S1 Fig. SDS-PAGE of purified recombinant CTSL.** M: Protein molecular weight marker; lane 1, total protein extracted from *E. coli* BL21 (DE3); lane 2: *E. coli* containing pET-24a-CTSL without IPTG induction; lane 3: *E. coli* containing pET-24a-CTSL with IPTG induction; lane 4: soluble protein; lane 5: inclusion bodies; lane 6: purified recombinant CTSL protein.
(TIF)

**S2 Fig. Titer determination of anti-CTSL polyclonal antibodies by ELISA.** After the final immunization, antiserum was serially diluted, and the absorbance values were measured.
(TIF)

## Author Contributions

**Conceptualization:** Yun-Ru Chiang, Han-Tso Lin, John Han-You Lin.

**Data curation:** Chao-Wei Chang.

**Investigation:** Yun-Ru Chiang, Chao-Wei Chang.

**Methodology:** Yun-Ru Chiang, Han-Tso Lin, Chao-Wei Chang, Shih-Ming Lin, John Han-You Lin.

**Software:** Yun-Ru Chiang.

**Supervision:** Shih-Ming Lin, John Han-You Lin.

**Writing – original draft:** Yun-Ru Chiang.

**Writing – review & editing:** John Han-You Lin.

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
