## [Decision Letter · Decision Letter 0]

20 Nov 2023

PONE-D-23-33754Dynamic expression of cathepsin L in the black soldier fly (Hermetia illucens) gut during Escherichia coli challengePLOS ONE

Dear Dr. Lin,

Thank you for submitting your manuscript to PLOS ONE. After careful consideration, we feel that it has merit but does not fully meet PLOS ONE’s publication criteria as it currently stands. Therefore, we invite you to submit a revised version of the manuscript that addresses the points raised during the review process.

We look forward to receiving your revised manuscript.

Kind regards,

Patrizia Falabella

Academic Editor

PLOS ONE

Journal Requirements:

Reviewers' comments:

Reviewer's Responses to Questions

**Comments to the Author**

1. Is the manuscript technically sound, and do the data support the conclusions?

Reviewer #1: Yes

Reviewer #2: Yes

2. Has the statistical analysis been performed appropriately and rigorously? 

Reviewer #1: Yes

Reviewer #2: Yes

3. Have the authors made all data underlying the findings in their manuscript fully available?

Reviewer #1: Yes

Reviewer #2: Yes

4. Is the manuscript presented in an intelligible fashion and written in standard English?

Reviewer #1: Yes

Reviewer #2: Yes

5. Review Comments to the Author

Reviewer #1: Line 41, please add references (for example https://doi.org/10.1111/1744-7917.13155, Bulletin of Insectology 75 (1): 75-82, 2022)

Line 63, please add more recent references (for example https://doi.org/10.3390/insects14050464)

Line 73, please insert a reference

MATERIAL AND METHODS

Line 120-125 Description of BSF is useless

Line 186: it is necessary to add the concentrations used

Line 190: it is necessary to add the concentrations used

Reviewer #2: Chiang et al describe expression of cathepsin L in black soldier fly

larvae (BSFL) while challenged by infection of E. coli.

The overall conclusion that under infection levels of CTSL are elevated

and that this provides support for the potential role of CTSL in

immunological response of BSFL in the gut are supported by

results. Therefore the ms is to be published. There are however a few

concerns, which should be addressed prior acceptance.

The RNA and immune analysis are consistent and refer to the specific

protein, called CTSL. This however can not be claimed for the

measurements of proteolytic activity.

The Z-Phe-Arg-AMC substrate used is not as specific for cathepsin L as

the authors believe. It can be used on other cathepsin as

well. Furthermore, this substrate was developed to be used on

mammalian cathepsins. As their sequence analysis shows the BSF CTLS

has 50% identity with human CTSL. The indicated difference already

indicates that the substrate is not as specific. Therefore I suggest

the authors to comment the following issues:

- how similar is BSF CTSL from other human cysteine cathepsins? Is the

human CTSL indeed its closest human homologue?

- Is CTSL the only cathepsin-like enzyme in the BSF genome? Other

species have multiple cathepsin. How much is known about this?

Namely, the kinetic data could have recorded also activities or

other cysteine cathepsins, which implies that also levels of other

cysteine cathepsins are elevated during the challenge.

- It is not obvious to me the relationship between the needle injury

and immune response, as the observed CTSL was mostly elevated in

guts tissue, which could have come in contact with E.coli also

during the feeding process. Please elaborate this.

6. PLOS authors have the option to publish the peer review history of their article (what does this mean?). If published, this will include your full peer review and any attached files.

Reviewer #1: No

Reviewer #2: **Yes: **Dusan Turk

---

## [Author Response · Author response to Decision Letter 0]

13 Dec 2023

We have responded to the reviewers' comments in the document titled 'Response to Reviewers.

---

## [Decision Letter · Decision Letter 1]

23 Jan 2024

Dynamic expression of cathepsin L in the black soldier fly (Hermetia illucens) gut during Escherichia coli challenge

PONE-D-23-33754R1

Dear Dr. Lin,

We’re pleased to inform you that your manuscript has been judged scientifically suitable for publication and will be formally accepted for publication once it meets all outstanding technical requirements.

Kind regards,

Patrizia Falabella

Academic Editor

PLOS ONE

Additional Editor Comments (optional):

Reviewers' comments:

Reviewer's Responses to Questions

**Comments to the Author**

1. If the authors have adequately addressed your comments raised in a previous round of review and you feel that this manuscript is now acceptable for publication, you may indicate that here to bypass the “Comments to the Author” section, enter your conflict of interest statement in the “Confidential to Editor” section, and submit your "Accept" recommendation.

Reviewer #1: All comments have been addressed

2. Is the manuscript technically sound, and do the data support the conclusions?

Reviewer #1: Yes

3. Has the statistical analysis been performed appropriately and rigorously? 

Reviewer #1: Yes

4. Have the authors made all data underlying the findings in their manuscript fully available?

Reviewer #1: Yes

5. Is the manuscript presented in an intelligible fashion and written in standard English?

Reviewer #1: Yes

6. Review Comments to the Author

Reviewer #1: I thank the authors for having addressed all the requested comment and suggestions.

The paper is complete.

7. PLOS authors have the option to publish the peer review history of their article (what does this mean?). If published, this will include your full peer review and any attached files.

Reviewer #1: No

---

## [Editor Report · Acceptance letter]

20 Feb 2024

PONE-D-23-33754R1 

PLOS ONE

Dear Dr. Lin, 

I'm pleased to inform you that your manuscript has been deemed suitable for publication in PLOS ONE. Congratulations! Your manuscript is now being handed over to our production team.

Kind regards, 

on behalf of

Prof. Patrizia Falabella 

Academic Editor

PLOS ONE